# Quantitative ultrasound, elastography, and machine learning for assessment of steatosis, inflammation, and fibrosis in chronic liver disease

**François Destrempes[1], Marc Gesnik[1], Boris Chayer[1], Marie-Hélène Roy-Cardinal[1], Damien Olivié[2,3], Jeanne-Marie Giard[4], Giada Sebastiani[5], Bich N. Nguyen[6,7], Guy Cloutier[1,2,8]⊙\*, An Tang[2,3,9]⊙\***

**1** Laboratory of Biorheology and Medical Ultrasonics, University of Montreal Hospital Research Center (CRCHUM), Montréal, Québec, Canada, **2** Department of Radiology, Radiation oncology and Nuclear Medicine, Université de Montréal, Montréal, Québec, Canada, **3** Department of Radiology, Centre hospitalier de l'Université de Montréal (CHUM), Montréal, Québec, Canada, **4** Department of Medicine, Division of Hepatology and Liver Transplantation, Université de Montréal, Montréal, Québec, Canada, **5** Department of Medicine, Division of Gastroenterology and Hepatology, McGill University Health Centre (MUHC), Montréal, Québec, Canada, **6** Department of Pathology, Centre hospitalier de l'Université de Montréal (CHUM), Montréal, Québec, Canada, **7** Department of Pathology and Cellular Biology, Université de Montréal, Montréal, Québec, Canada, **8** Institute of Biomedical Engineering, University of Montreal, Montréal, Québec, Canada, **9** Laboratory of Medical Image Analysis, Centre de recherche du Centre hospitalier de l'Université de Montréal (CRCHUM), Montréal, Québec, Canada

⊙ These authors contributed equally to this work.
\* guy.cloutier@umontreal.ca (GC); an.tang@umontreal.ca (AT)

**Data Availability Statement:** All relevant data are within the paper and its Supporting Information files.

## Abstract

### Objective

To develop a quantitative ultrasound (QUS)- and elastography-based model to improve classification of steatosis grade, inflammation grade, and fibrosis stage in patients with chronic liver disease in comparison with shear wave elastography alone, using histopathology as the reference standard.

### Methods

This ancillary study to a prospective institutional review-board approved study included 82 patients with non-alcoholic fatty liver disease, chronic hepatitis B or C virus, or autoimmune hepatitis. Elastography measurements, homodyned K-distribution parametric maps, and total attenuation coefficient slope were recorded. Random forests classification and boot-strapping were used to identify combinations of parameters that provided the highest diagnostic accuracy. Receiver operating characteristic (ROC) curves were computed.

### Results

For classification of steatosis grade S0 vs. S1-3, S0-1 vs. S2-3, S0-2 vs. S3, area under the receiver operating characteristic curve (AUC) were respectively 0.60, 0.63, and 0.62 with elasticity alone, and 0.90, 0.81, and 0.78 with the best tested model combining QUS and

**Funding:** This work was supported by grants from the Canadian Institutes of Health Research (CIHR)-Institute of Nutrition, Metabolism and Diabetes (INMD) (CIHR-INMD #273738 and #301520, https://cihr-irsc.gc.ca/) to AT. This work was also supported by Junior 1 and Junior 2 Clinical Research Scholarships from the Fonds de recherche du Québec en Santé (FRQS, https://frq.gouv.qc.ca/en/) and Fondation de l'Association des radiologistes du Québec (FARQ) (FRQS-FARQ #26993 and #34939 to AT; and by a Junior 1 Clinical Research Scholarships from FRQS (#27127) and research salary from McGill University to GS. The funders had no role in study design, data collection and analysis, decision to publish, or preparation of the manuscript.

**Competing interests:** The authors have declared that no competing interests exist.

**Abbreviations:** ACS, attenuation coefficient slope; AUC, area under the receiver operating characteristic curve; CLD, chronic liver disease; CRN, Clinical Research Network; HKD, homodyned K-distribution; IQR, inter-quartile range; MNTN, maximum number of terminal nodes; NAFLD, nonalcoholic fatty liver disease; NASH, nonalcoholic steatohepatitis; pSWE, point shear wave elastography; QUS, quantitative ultrasound; ROC, receiver operating characteristic; ROI, region of interest.

elastography features. For classification of inflammation grade A0 vs. A1-3, A0-1 vs. A2-3, A0-2 vs. A3, AUCs were respectively 0.56, 0.62, and 0.64 with elasticity alone, and 0.75, 0.68, and 0.69 with the best model. For classification of liver fibrosis stage F0 vs. F1-4, F0-1 vs. F2-4, F0-2 vs. F3-4, F0-3 vs. F4, AUCs were respectively 0.66, 0.77, 0.72, and 0.74 with elasticity alone, and 0.72, 0.77, 0.77, and 0.75 with the best model.

## Conclusion

Random forest models incorporating QUS and shear wave elastography increased the classification accuracy of liver steatosis, inflammation, and fibrosis when compared to shear wave elastography alone.

## Introduction

Chronic liver disease (CLD) is one of the top ten leading causes of death in the United States [1]. Nonalcoholic fatty liver disease (NAFLD) is the most common cause of CLD, affecting up to one third of the adult Western population [2]. It has a substantial burden due to the incidence and prevalence, and its impact on longevity and quality of life [3]. NAFLD is characterized by vacuoles of fat and may lead to nonalcoholic steatohepatitis (NASH), which is characterized by inflammation. All causes of CLD may evolve to liver fibrosis, a scarring process, which may progress to cirrhosis and liver failure. Although liver biopsy is the established reference standard for classification of steatosis grade, inflammation grade, and fibrosis stage, it has several limitations including cost, sampling error, and procedure-related morbidity and mortality [4].

The noninvasiveness, wide availability, innovative technical developments, and cost-effectiveness of ultrasound constitute key advantages for management of patients with CLD. Medical ultrasound has traditionally been used to diagnose diseases based on B-mode (structures) and Doppler-mode (flow) images. More recently, elastography techniques, which assess stiffness, tissue strain or viscoelasticity, provide the highest diagnostic performance for staging of liver fibrosis [5–7]. Over the past 5 decades, quantitative ultrasound (QUS) imaging techniques have been developed to analyze constructive and destructive interferences between echoes to characterize tissue microstructure below the typical resolution of ultrasound scanners, based on spectral analysis of radiofrequency signals or statistical properties of the echo envelope [8]. In earlier clinical studies, QUS controlled attenuation [9–11], which is related to ultrasound energy loss in tissues, and backscatter coefficient [12] provided promising results for grading liver steatosis. More recently, several studies have proposed traditional machine learning [13–16] or deep learning [17–19] techniques combined with shear wave elastography to improve the staging of liver fibrosis. The homodyned-K distribution has also been proposed to model the ultrasound echo envelope in the context of liver steatosis and steatohepatitis [12, 20–22], since its parameters are related to cell density, tissue echogenicity, and tissue microstructure.

Considering the high disease prevalence, there is a need for noninvasive imaging and classification of CLD. The purpose of this study was to develop a combined QUS and elastography-based model to improve classification of steatosis grade, inflammation grade, and fibrosis stage in comparison with elastography alone, using histopathology as the reference standard. We privileged the random forests machine learning strategy to identify combination of imaging features providing the highest classification accuracy for a given training dataset [21].

## Materials and methods

### Study design and subjects

This is a retrospective, cross-sectional ancillary study to an imaging trial (ClinicalTrials.gov Identifier No. NCT02044523). This study was approved by the institutional review board of the two participating institutions, Centre hospitalier de l'Université de Montréal and McGill University Health Centre. All patients gave written informed consent.

The prior cross-sectional imaging trial compared diagnostic accuracy of ultrasound-based and magnetic resonance-based elastography techniques using liver biopsy as the reference standard [6]. For this ancillary study, QUS and elastography measurements were used to develop models for classification of steatosis grade, inflammation grade, and fibrosis stage.

Between October 2014 and September 2018, consecutive adult subjects were enrolled if a liver biopsy was scheduled as part of their clinical standard of care. Ninety-one subjects recruited at hepatology clinics of the two participating institutions underwent QUS. Subjects were excluded from this ancillary study if: (a) images were not acquired with the probe assigned to the study protocol ($n = 2$); (b) the underlying pathology did not meet eligibility criteria due to ethanol consumption ($n = 2$), drug-induced hepatitis ($n = 1$), sarcoidosis ($n = 1$), cholestasis ($n = 1$), or no biopsy was performed ($n = 2$). Characteristics of 82 subjects included in this study are described in Table 1. Among the 82 included subjects, the underlying pathology was CLD attributable to NAFLD ($n = 6$), NASH ($n = 38$), hepatitis B virus ($n = 2$), hepatitis C virus ($n = 13$), autoimmune hepatitis ($n = 16$), or mixed causes ($n = 7$). Consecutive eligible participants were included.

For each of steatosis or inflammation grade and fibrosis stage, classification tasks obtained from splitting the dataset into two classes based on the grade (4 values) or stage (5 values), viewed as ordinal variables, were assessed. Thus, we considered 3 steatosis and inflammation classification tasks, and 4 fibrosis classification tasks.

### Ultrasound imaging

Ultrasound images were acquired with a clinical scanner (Acuson S2000 or S3000, Siemens Healthineers) using a convex probe (4C1). Patients were required to fast at least 4 hours prior to the examination. They were scanned in dorsal decubitus with their right arm in maximal abduction. Required standard intercostal B-mode images included: (1) the right liver lobe and right kidney, (2) right liver lobe at the level of the right portal vein, and (3) hepatic veins. A 3-second cine-loop of radiofrequency (RF) signals of the right liver lobe in a plane without major vessels was recorded for subsequent QUS post-processing.

### Ultrasound shear wave elastography

Point shear wave elastography (pSWE) was performed with the same convex probe according to guidelines [23]. The median shear wave velocity (expressed in m/s) of 10 valid measurements (obtained in 20 repetitions or less) was used as a surrogate biomarker of liver stiffness. Reliability of measurements was based on the success rate and an interquartile range to the median (IQR/M) < 30% as per European Federation of Societies for Ultrasound in Medicine and Biology (EFSUMB) guidelines and Recommendations on the Clinical Use of Liver Ultrasound Elastography [23].

### Preliminary post-processing

Echo envelope of RF data was computed based on Hilbert's transform, after having compensated for time gain compensation settings. The first 30 images of uncompressed and unfiltered

**Table 1. Characteristics in 82 patients.**

| Characteristic | Data |
|---|---:|
| Sex | |
| Male | 42 (51%) |
| Female | 40 (49%) |
| Age (y) | |
| Mean ± SD (range) | 56 ± 12 (23–78) |
| BMI (kg/m$^2$) | |
| Mean ± SD (range) | 30.0 ± 5.8 (17–45) |
| < 25 | 16 (20%) |
| ≥ 25 and < 30 | 23 (28%) |
| ≥ 30 and < 40 | 39 (47%) |
| ≥ 40 | 4 (5%) |
| Racial category | |
| White | 62 (76%) |
| Black | 4 (5%) |
| Asian | 2 (2%) |
| American Indian | 2 (2%) |
| Hawaiian or Pacific Islander | 1 (1%) |
| N/A | 11 (14%) |
| Diabetes | 27 (33%) |
| Hypertension | 32 (39%) |
| Laboratory tests: Mean ± SD (range) | |
| AST (U/L) | 56 ± 55 (14–319) |
| ALT (U/L) | 75 ± 81 (13–473) |
| GGT (U/L) | 77 ± 93 (10–464) |
| Platelet count (x 10$^9$/L) | 201 ± 66 (87–383) |
| Total bilirubin (μmol/L) | 12.5 ± 5.0 (4.5–28.5) |
| Prothrombin time (%) | 99.3 ± 7.8 (83–120) |
| Alkaline phosphatase (U/L) | 76 ± 36 (32–217) |
| Albumin (g/L) | 41.2 ± 6.4 (31–79) |
| Cholesterol (mmol/L) | 4.6 ± 1.0 (2.9–7.0) |
| Biopsy length (mm) | |
| Mean ± SD (range) | 20.1 ± 5.1 (10–30) |
| Fibrosis stage | |
| F0 (none) | 12 (14%) |
| F1 (perisinusoidal or periportal) | 13 (16%) |
| F2 (periportal and presence of septa) | 18 (22%) |
| F3 (numerous septa without cirrhosis) | 13 (16%) |
| F4 (cirrhosis) | 26 (32%) |
| Inflammation activity grade | |
| A0 (none) | 8 (10%) |
| A1 (negligible) | 39 (47%) |
| A2 (moderate) | 27 (33%) |
| A3 (severe) | 8 (10%) |
| Steatosis grade | |
| S0 (<5% hepatocytes involved) | 29 (35%) |
| S1 (5%-33% hepatocytes involved) | 22 (27%) |
| S2 (33%-66% hepatocytes involved) | 15 (18%) |

*(Continued)*

**Table 1.** (Continued)

| Characteristic | Data |
|---|---|
| S3 (>66% hepatocytes involved) | 16 (20%) |
| Iron | |
| 0 | 59 (72%) |
| 1 | 13 (16%) |
| 2 | 2 (2%) |
| 3 | 0 (0%) |
| 4 | 0 (0%) |
| N/A | 8 (10%) |

Numbers in parentheses are percentages, unless otherwise specified.

SD = standard deviation, BMI = body mass index, AST = aspartate aminotransferase, ALT = alanine
aminotransferase, GGT = gamma-glutamyl transpeptidase.

echo envelopes of RF signals were analyzed. A region of interest (ROI) was manually delineated in the first image of the cineloop and then propagated automatically to subsequent images. A pre-classification of pixels (pixels' labeling) within the ROI of all frames was estimated [24], according to the statistical distribution of the echo envelope.

## Quantitative ultrasound analysis

A sliding window of 78 x 12 pixels, corresponding to about 3 mm along both axial and lateral axes, was swept across the ROI by steps of 4 x 1 pixels. For each window, only pixels with same label as the central pixel were used for homodyned K-distribution (HKD) parameters' estimation [24], so that the hypothesis of a single distribution was met locally (as opposed to a mixture of a few distributions), thus allowing for application of the XU estimation method of the HKD parameters [25]. Notice that the HK distribution was proposed as a general model for echo envelope distribution, with the advantage of a physical interpretation of its parameters [24]. These QUS maps comprised the normalized mean intensity $\mu_n$ (mean intensity divided by the maximal intensity within ROI) [26]; the reciprocal $1/\alpha$ of the scatterer clustering parameter $\alpha$, which is an indicator of density and homogeneity in fluctuations of acoustical impedance [1]; the coherent-to-diffuse signal ratio $k$ [27] and the diffuse-to-total signal power ratio $1/(k+1)$ [1], both of which are QUS biomarkers of structure within scatterers' spatial organization [24]. Notice that $k$ represents the coherent-to-diffuse signal power ratio, not to be confused with parameter $k$ above. For each parametric map, mean value and inter-quartile range (IQR) within the ROI were computed, and median values over the 30 frames yielded two QUS features per map [21]. Thus, 8 HKD features were produced for each acquisition: (mean and IQR features) x (4 parametric maps). Means and maxima were winsorized to avoid spurious outliers [28].

The total acoustical attenuation due to layers of fat that could be present in patients with obesity might be a confounder for HKD parameters. Therefore, the total attenuation coefficient slope (ACS) was estimated, based on spectral Gaussian fit method [8], and its median value over 30 frames was output as an QUS feature, in addition to HKD parameters. Combination of total attenuation with statistical features was shown to improve classification performance [29], and this is an alternative to estimating HKD parameters directly on the echo envelope compensated for total attenuation.

Lastly, the local ACS was estimated as a further QUS biomarker. Estimation was based on a hybrid state-of-the-art method [30]. For both total and local attenuation estimation methods,

a calibrated reference phantom (model 117GU-101, CIRS, Norfolk, VA) was used to compensate for acquisition settings [8]. In total, 11 features were extracted from each acquisition of radiofrequency data.

B-mode images with overlaid parametric maps of QUS features in 4 representative patients are shown in Fig 1. Point shear wave elastography images in the same patients are shown in Fig 2. Custom programs for QUS computations were implemented in C++ language and Matlab R2018a software (MathWorks).

**Histopathology analysis.** Liver biopsies were performed by percutaneous approach with 16-G or 18-G core needles. Specimens were stained with hematoxylin and eosin and centrally scored by a hepatopathologist. Steatosis was graded according to the proportion of hepatocytes with macrovesicles of fat according to the NASH Clinical Research Network (CRN) scoring system, where S0: < 5%, S1: 5–33%, S2: 33–66%, and S3 > 66%. Inflammation was graded according to the level of lobular inflammation with the NASH CRN system for patients with NAFLD or NASH, and according to the severity of inflammation with the METAVIR scoring system for patients with autoimmune or chronic hepatitis, where A0: no foci, A1: < 2 foci per 200x field, A2: 2–4 foci per 200x field, and A3: > 4 foci per 200x field. Fibrosis was staged according to the distribution and severity of fibrosis and level of architectural modeling with the METAVIR scoring system for patients with autoimmune hepatitis or hepatitis B or C infection, and according to the NASH CRN for patients with NAFLD or NASH. Fibrosis stages F1A, F1B, and F1C in the NASH CRN system were pooled into stage F1.

## Machine learning model and features selection

Classification tasks were performed with random forests [31]. This statistical learning approach is suitable in the case of relatively small data sets, as random forests avoid over-fitting (*i.e.*, that would be based on too many trees—see Theorem 1.2 and the remark thereafter in [31]), and have very few hyper-parameters to be tuned; *i.e.*, input features and maximum number of terminal nodes (MNTNs) in each tree. For classification purposes, we used 1000 trees and let MNTN vary from 2 to 20 by steps of 2. Selection of features was performed with G-mean as a recommended evaluation index in the case of imbalanced data set [32]. Random forests used for feature selection comprised 3000 trees. The 10 combinations of 4 features or less among all 11 features that obtained highest G-mean were selected, together with any other combination with same G-mean values in case of ties. Notice that random forests do not perform pruning on trees [33] and that Gini index is used at each node to determine the best split [34]. The number of variables randomly selected at each node is the default one; i.e., the square root of the total number of variables (one or two candidates per node in the context of this study since we used one to four features for classification).

Features selection was performed by testing all possible combinations of 1 to 4 features (there are 561 such combinations) as follows. Each combination of features was considered as the input of its own random forest consisting of 3000 trees. A random forest was trained for each of these combinations of features on the whole dataset. As a measure of potential quality of a given combination of features that was quick to compute (unlike an AUC obtained from several bootstraps), the G-mean was computed based on the sensitivity and specificity obtained on the whole dataset. The 10 combinations of 4 features or less among all 11 features that obtained highest G-mean were selected, together with any other combination with same G-mean values in case of ties. Further assessment of the 10 (or more) selected combinations of features was then performed based on AUCs estimated with the 0.638+ bootstrap method. See S1 Table for AUCs obtained with 0.638+ bootstrap method

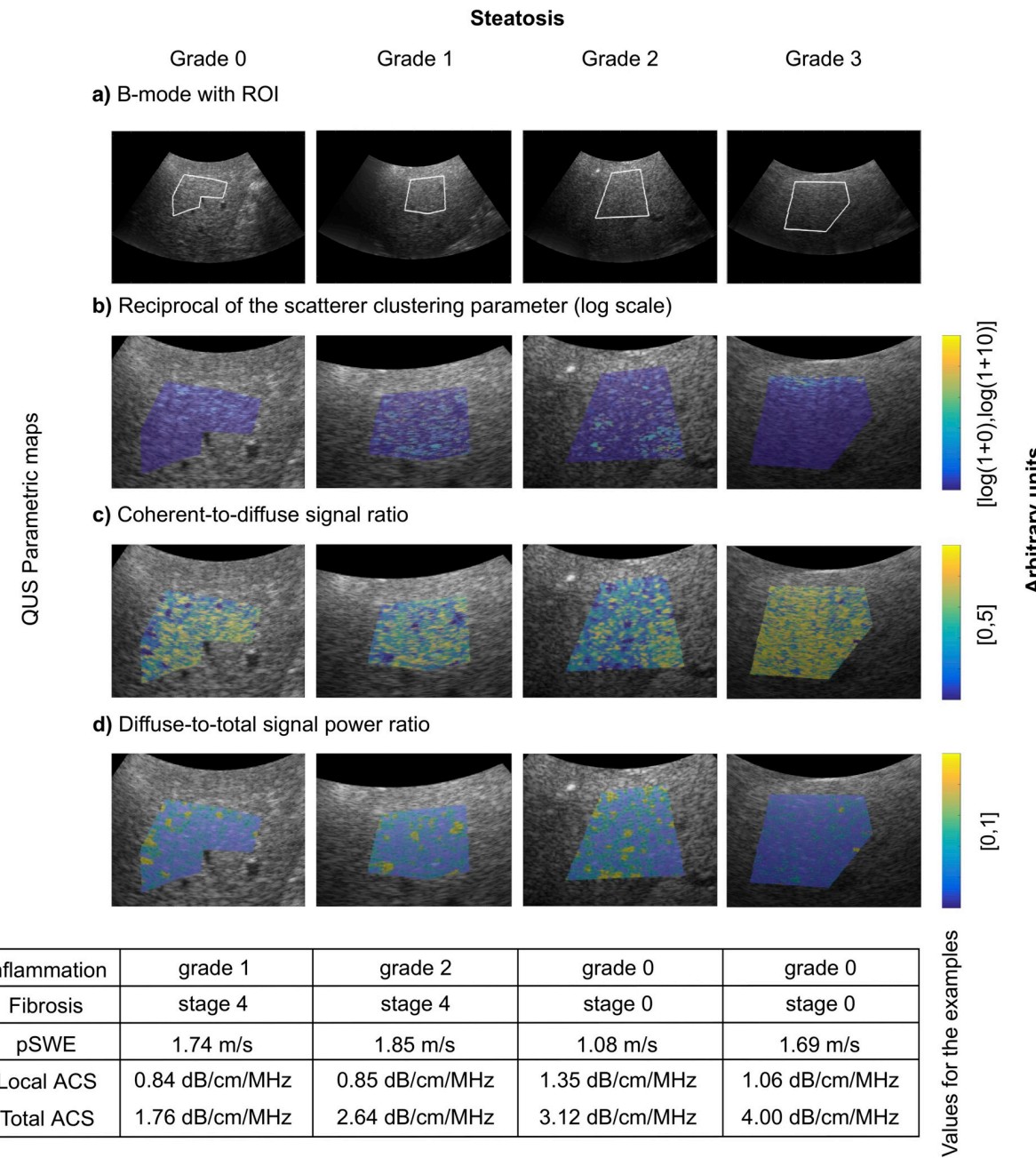

**Fig 1. Examples of histopathology-proven steatosis grades.** From left to right with histopathology-proven steatosis grades: 1) 62-year-old man with hepatitis B virus with steatosis grade 0; 2) 62-year-old man with NASH and hepatitis B virus with steatosis grade S1; 3) 38-year-old man with NASH with steatosis grade S2; and 4) 69-year-old woman with NASH with steatosis grade S3. Displayed images: (A) Representative B-mode acquired in the right liver lobe and corresponding QUS parametric maps (zoomed in on ROIs) illustrating (B) reciprocal of scatterer clustering parameter (log scale), (C) coherent-to-diffuse signal ratio, and (D) diffuse-to-total signal power ratio. Yellow indicates higher values, whilst dark blue values indicate lower values. In the table are reported the inflammation grade and the fibrosis stage, as well as the corresponding point shear wave elasticity (pSWE), and local and total attenuation coefficient slopes (ACS).

for each of the 11 features taken individually. For inflammation grade 0 vs. $\geq$ 1, the AUC of total attenuation was within the AUC's 95% CI corresponding to the best tested combination of features. For inflammation grade A0-1 vs. A2- = 3, the AUC of parameter $k$ was

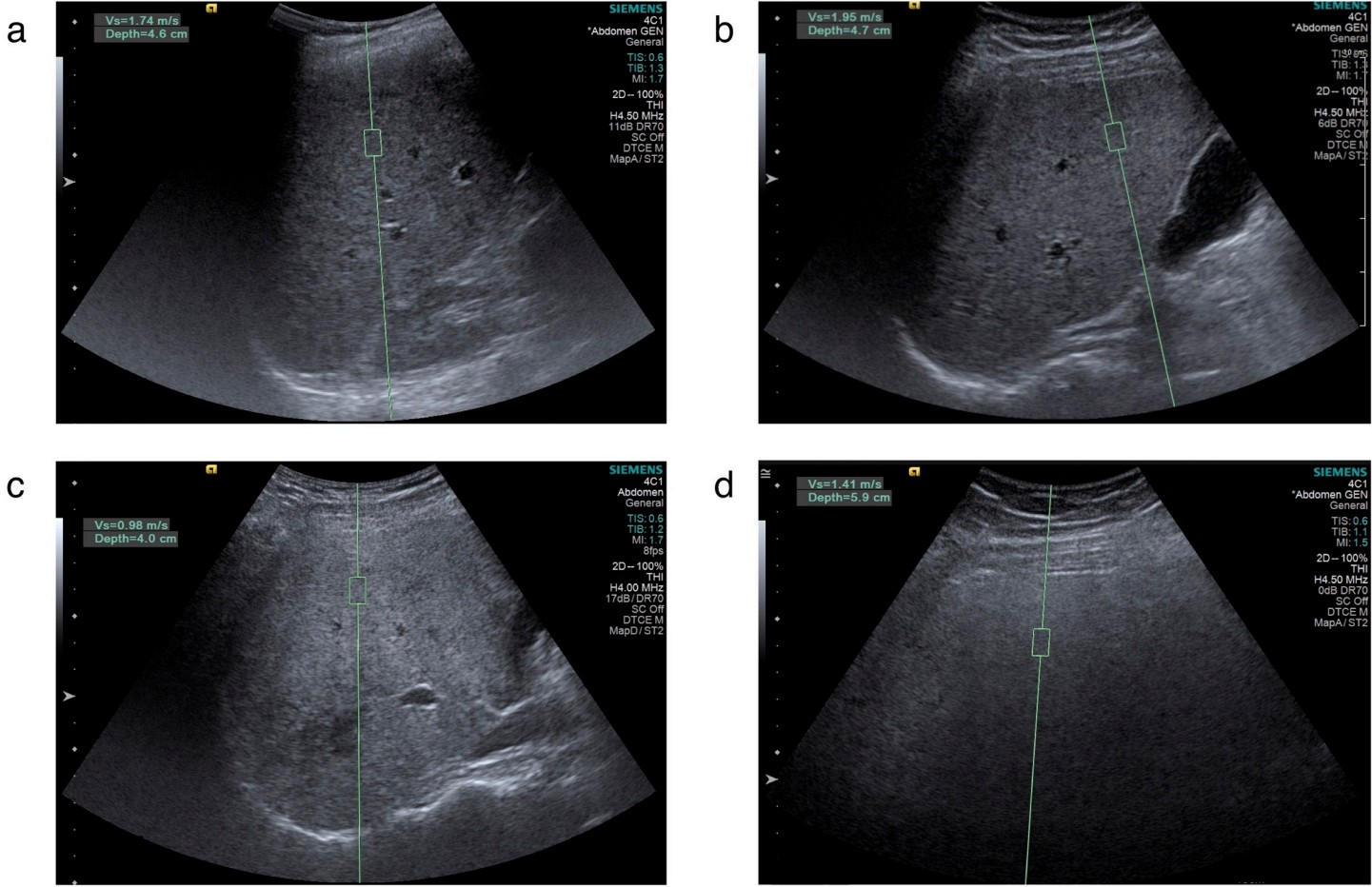

**Fig 2. Examples of point shear wave elastography (pSWE) measurements.** pSWE measurements in 4 different patients: (A) steatosis grade S0, (B) steatosis grade S1, (C) steatosis grade S2, and (D) steatosis grade S3. Green rectangles indicate regions of interest where measurements are performed.

within the corresponding AUC's 95% CI. For fibrosis stage F0 vs. F1-4, F0-1 vs. F2-4, or F0-3 vs. 4, the AUCs of parameter pSWE were within the corresponding 95% CIs corresponding to the best combinations of features. For all other classification tasks, none of the best performing individual features were within the corresponding AUC's 95% CI of the best tested combination of features. S2 Table presents ANOVA tests and post-hoc multiple comparisons when applicable for each of these features. AUCs obtained with 0.638+ bootstrap method in the case where all features were taken as input of random forests are presented in S3 Table. The resulting AUCs all fell off below the AUC's 95% CI of the corresponding best tested combinations of features.

ROC curves were obtained using stratified sampling, akin to varying the loss function used at the learning phase according to the sensitivity aimed at [35]. This may explain the non-monotonicity appearing in some of the reported ROC curves, as the dataset was imbalanced for some of the dichotomous classification tasks. We did not apply post-processing algorithms on the reported ROC curves.

## Evaluation of classification performance

Classification performance was evaluated with the 0.632+ bootstrap method [36]. This method, which is recommended when sample size is too small for allowing training-

validation-test set split, combines the leave-one-out bootstrap error and the training error with adaptive weights, thus avoiding over-estimation of the generalization error by the former and under-estimation by the latter. Receiver operating characteristic (ROC) curves were estimated by considering stratified resampling of the dataset with proportion of one class varying from 0 to 1 by steps of 1/40. One thousand bootstraps were generated for each stratification proportion, and for each bootstrap, classification errors were evaluated only on data not belonging to the bootstrap. The area under receiver operating characteristic curves (AUC-ROC) were then computed with the trapezoidal method for each previously selected combination of features and each value of MNTN. Combination of features and corresponding MNTN with highest AUC-ROC was selected as best model among tested ones for a given classification task. The jackknife method [37] was used to generate a sample of ROC-AUCs, from which a 95% confidence interval (CI) could be estimated based on percentiles. For each classification task, the point with maximal Youden index on the ROC curve of its best model was computed, and corresponding sensitivity and specificity were reported. The flow chart of the full post-processing pipeline is presented in Fig 3.

## Blinding

Sonographers performing ultrasound examinations were blinded to histopathological results. The hepatopathologist was blinded to ultrasound results.

## Statistical analysis

To evaluate relationships between single features and liver steatosis, inflammation, and fibrosis, features were further analyzed with univariate ordinal logistic regressions. Steatosis grade, inflammation grade, and fibrosis stage were considered in turn as response, and any of the features appearing within the best combinations for the corresponding classification task was considered as single predictor. Coefficients of these regressions were reported, together with *P*-values, after Holm-Bonferroni correction.

To test for association between histopathological components (steatosis, inflammation, and fibrosis), viewed as categorical variables, contingency tables for pairs of these variables were computed, and Pearson's chi-square tests were performed based on corresponding tables. *P*-values were reported, after Holm-Bonferroni correction.

Statistical analyses were performed with software *R* (version 3.2.5, 2016; *R* Foundation for Statistical Computing). For proportional odds logistic regressions (also called ordinal logistic regressions), we have used the function "polr" of the package "MASS" [38, 39] usable with the software *R*. The raw p-values were obtained with the function "pnorm" based on the statistics *t* output by the function "polr". The Holm-Bonferroni correction on p-values was then computed according to the algorithm introduced in [40]. Machine learning and regression analyses used the package "randomForest" (version 4.6–12, 2015) [41] and "MASS" (version 7.3–45, 2015) [42], respectively.

## Results

### Study database

Table 1 indicates the sample size of each group determined by histopathology: 29, 22, 15, and 16 patients for steatosis grades S0, S1, S2 and S3, respectively; 8, 39, 27, and 8 patients for inflammation grades A0, A1, A2 and A3, respectively; and 12, 13, 18, 13, and 26 patients for fibrosis stages F0, F1, F2, F3, and F4, respectively. Examples of representative histopathology images are presented in Fig 4. Dataset is provided in S4 Table.

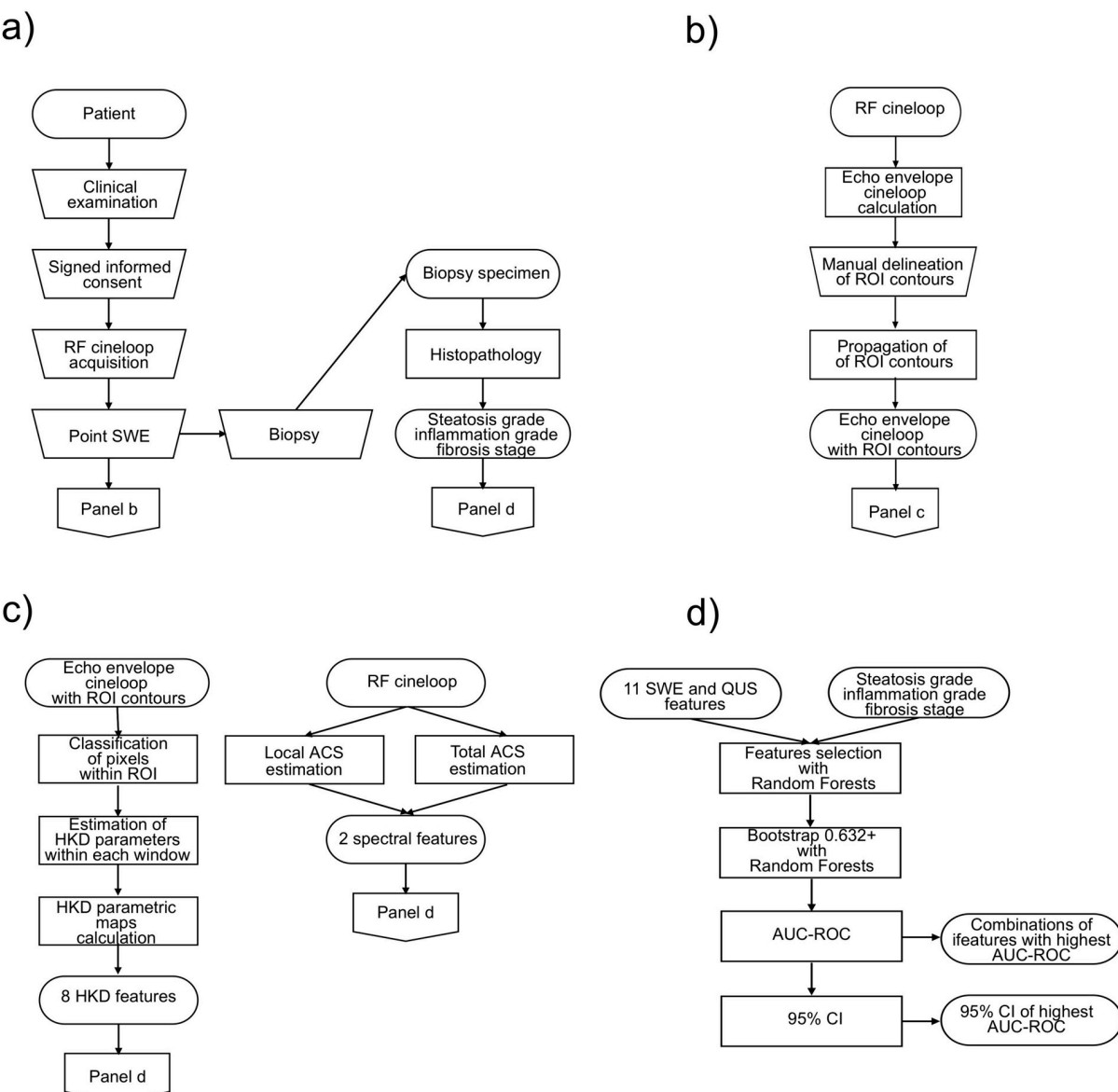

**Fig 3. Flow chart of the post-processing pipeline.** (A) Clinical examination, recruitment with signed informed consent, research data acquisitions, biopsy and histology. (B) Calculation of the echo envelope of radiofrequency (RF) cineloops, manual delineation of a region-of-interest (ROI) contours in one frame, and propagation of contours along all frames of the cineloop. (C) Calculation of quantitative ultrasound (QUS) features. (D) Machine learning with random forests based on 11 features and gold standards: selection of 10 combinations of 4 features or less with highest *G*-mean (features selection), calculation of AUC-ROC on these combinations (0.632+ bootstrap), and estimation of the 95% CI for the features combination with highest AUC-ROC.

### Machine learning model and features selection

In Table 2, combination of QUS features and/or elasticity providing the highest AUC-ROC is reported for each classification task. AUCs obtained with elasticity alone were improved by combining QUS and elastography features, for each classification task, except for fibrosis stage $\leq 1$ vs. $\geq 2$. Improvements were most substantial for steatosis grade (25%-50% in AUC) and for classification of inflammation 0 vs. $\geq 1$ (34% in AUC). For the six other dichotomous

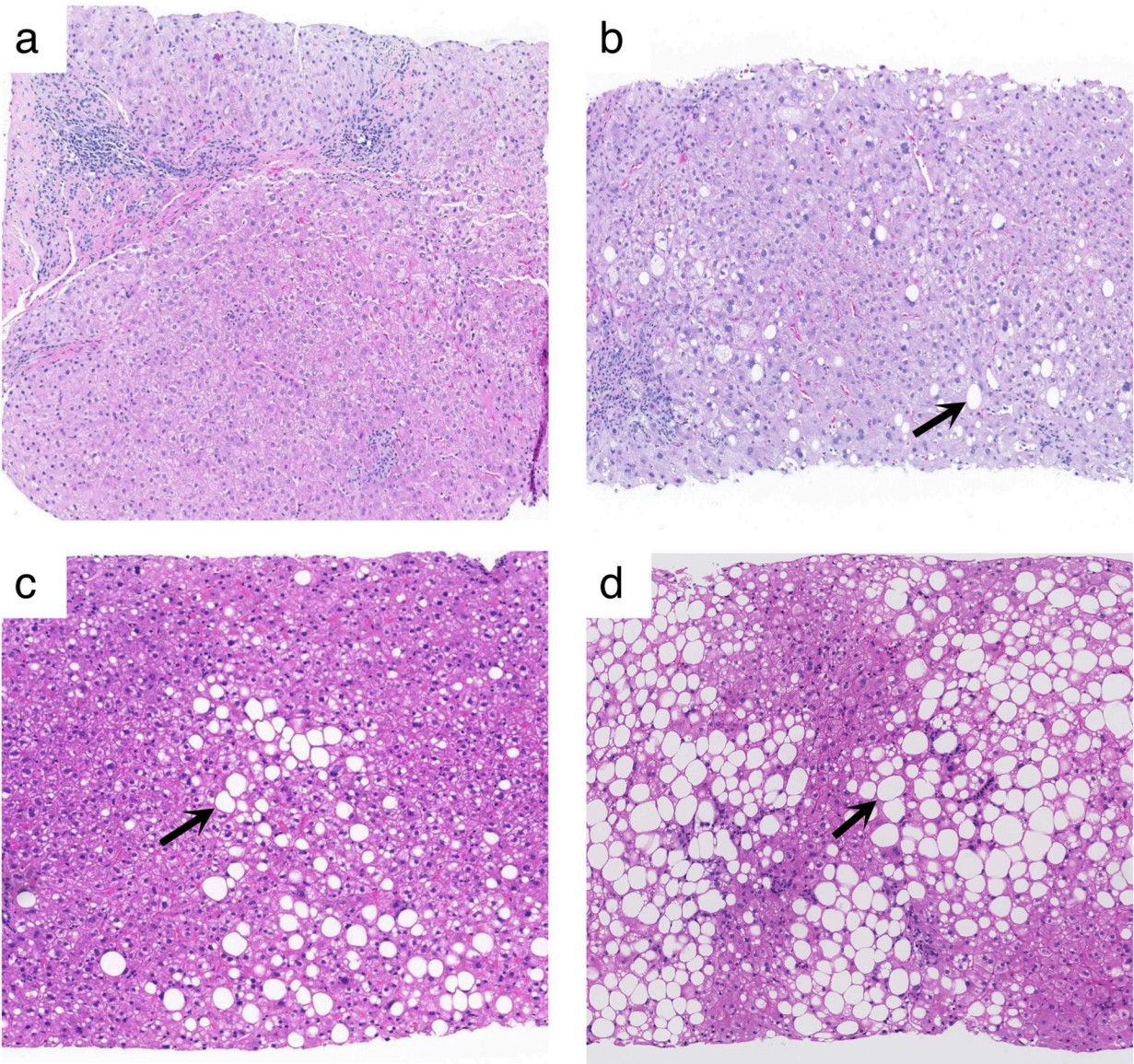

**Fig 4. Examples of histopathology slides.** Hematoxylin and eosin-stained images (10x magnification) in 4 different patients corresponding to those shown in Fig 1 above: (A) steatosis grade S0, (B) steatosis grade S1, (C) steatosis grade S2, and (D) steatosis grade S3. Representative vacuoles of macrovesicular steatosis are indicated by arrows.

tasks, improvement was less than 10% in AUC, despite the fact that for five of them, QUS features were combined with shear-wave elasticity.

## Classification tasks performance

Table 2 also provides AUC-ROC and its 95% confidence interval, for each classification task obtained with shear wave elasticity only, or with best combination of QUS and elastography features. For classification of steatosis grades S0 vs. S1-3, S0-1 vs. S2-3, S0-2 vs. S3, AUCs were 0.60, 0.63, 0.62, respectively, with elasticity alone, and were 0.90, 0.81, and 0.78, respectively, with the best tested combination of features. For classification of inflammation grades A0 vs. A1-3, A0-1 vs. A2-3, A0-2 vs. A3, AUCs were respectively 0.56, 0.62, and 0.64 with elasticity

**Table 2. Accuracy of shear wave elasticity alone and in combination with quantitative ultrasound (QUS) features for classification of steatosis, inflammation, and fibrosis.**

| Pathological features | Groups | Size | AUC-ROC | AUC-ROC | Parameters |
|---|---|---|---|---|---|
| | | | pSWE only | Multi-parameter | |
| Steatosis | S0 vs. S1-3 | 29/53 | 0.60 (0.59–0.61) | 0.90 (0.89–0.91) | $k$ IQR + $1/(k + 1)$ IQR + Local ACS |
| | S0-1 vs. S2-3 | 51/31 | 0.63 (0.62–0.66) | 0.81 (0.80–0.83) | $k$ IQR + $pSWE$ + Local ACS |
| | S0-2 vs. S3 | 66/16 | 0.62 (0.60–0.64) | 0.78 (0.77–0.79) | $k$ Mean + $\mu_n$ Mean + Local ACS |
| Inflammation | A0 vs. A1-3 | 8/74 | 0.56 (0.54–0.60) | 0.75 (0.73–0.76) | Total ACS |
| | A0-1 vs. A2-3 | 47/35 | 0.62 (0.61–0.64) | 0.68 (0.67–0.71) | $k$ Mean |
| | A0-2 vs. A3 | 74/8 | 0.64 (0.60–0.65) | 0.69 (0.66–0.71) | $pSWE$ + $1/\alpha$ IQR |
| Fibrosis | F0 vs. F1-4 | 12/70 | 0.66 (0.64–0.69) | 0.72 (0.69–0.74) | $pSWE$ + $k$ Mean +$1/(k + 1)$ Mean |
| | F0-1 vs. F2-4 | 25/57 | 0.77 (0.76–0.78) | 0.77 (0.76–0.80) | $pSWE$ + $\mu_n$ IQR + $k$ IQR + $1/(k + 1)$ IQR |
| | F0-2 vs. F3-4 | 43/39 | 0.72 (0.72–0.74) | 0.77 (0.76–0.79) | $pSWE$ + $\mu_n$ IQR +$1/(k + 1)$ IQR |
| | F0-3 vs. F4 | 56/26 | 0.74 (0.73–0.75) | 0.75 (0.74–0.77) | $pSWE$ + $\mu_n$ IQR +$1/(k + 1)$ IQR + Total ACS |

ACS = attenuation coefficient slope. AUC-ROC = area under the receiver operating characteristic curve. Numbers in parentheses are 95% confidence intervals. size = $N/M$, where $N$ = number of cases (out of 82 patients) such that pathological feature $\leq x$ (= 0, 1, 2, or 3) and $M$ = 82 –$N$; $pSWE$ = point shear wave elasticity; $\mu_n$ = mean intensity normalized by its maximal value; $1/\alpha$ = reciprocal of the scatterer clustering parameter; $k$ = coherent-to-diffuse signal ratio; $1/(k + 1)$ = diffuse-to-total signal power ratio; IQR = inter-quartile range.

alone, and raised up to 0.75, 0.68, and 0.69, respectively, with the best selected combination of features. For classification of liver fibrosis stages F0 vs. F1-4, F0-1 vs. F2-4, F0-2 vs. F3-4, F0-3 vs. F4, elasticity alone yielded AUCs of 0.66, 0.77, 0.72, and 0.74, respectively, whereas best tested combinations of features gave AUCs of 0.72, 0.77, 0.77, and 0.75.

From Table 2, one observes that local ACS is the only common parameter in the best combinations of parameters for the three dichotomous classifications tasks of steatosis grade. Moreover, the three dichotomous classification tasks of the inflammation grade have no common parameters. For the four dichotomous classification tasks of the fibrosis stage, point shear wave elasticity is the only common parameter. However, except for the task F0 vs. F1-4, for which the best combination of parameters was pSWE, $\mu_n$ mean and $1/(k + 1)$ mean, there were three common parameters: pSWE, $\mu_n$ IQR and $1/(k + 1)$ IQR.

The importance values, viewed as mean decrease in accuracy, of each of the parameters within the combinations of parameters appearing in Table 2 are reported in S5 Table, based on the R package "randomForest".

ROC curves for combination of elastography and QUS features with highest ROC-AUC for classification of steatosis, inflammation, and fibrosis are shown in Fig 5.

For sake of comparison, we present in S5 Table in Supplemental Materials the ROC-ACUs obtained with the support vector machine (SVM) model on the same combinations of parameters that are appearing in Table 2. For this purpose, we used the R package "e1071" (version 1.7–9, 2021) [43]. The kernel for projection of data in higher dimensional space was the radial basis function (RBF) of degree 3 with parameter "γ" equal to the reciprocal of the number of parameters (default values in the package). The ROC curves were produced with class weights of the form (p/N(0), (1-p)/N(1)), where N(0) and N(1) represent the number of elements in the two groups of a given dichotomous classification task, letting p vary from 0 to 1. As the cost $C$ related to the soft margin of SVM has an impact on accuracy, powers of 2 (with exponent varying from 0 to 10) were tested with the 0.632+ bootstrap method, and the best obtained values are reported in S6 Table. In cases where the best AUC was obtained with $C = 2^{10}$, we tested further powers of 2 until decrease in ACU. Finer tuning may have been

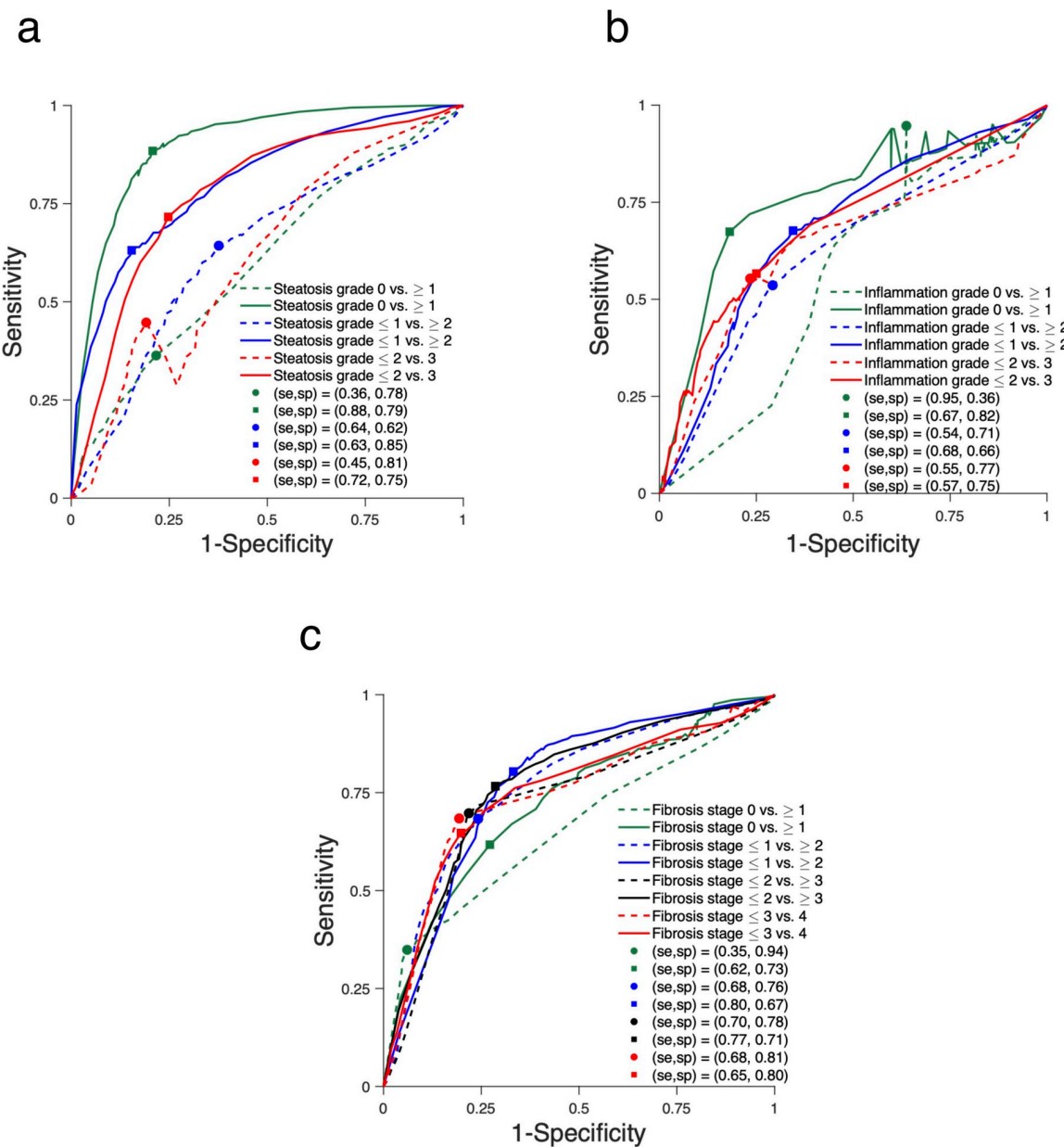

**Fig 5. Receiver operating characteristic (ROC) curves.** ROC curves obtained with elastography (dashed lines) and the combinations of QUS and elasticity features (solid lines) with highest AUC-ROC for the classification of (A) steatosis grade S0 vs. S1-3, S0-1 vs. S2-3, S0-2 vs. S3, (B) inflammation grade A0 vs. A1-3, A0-1 vs. A2-3, A0-2 vs. A3, and (C) fibrosis stage F0 vs. F1-4, F0-1 vs. F2-4, F0-2 vs. F3-4, F0-3 vs. F4. Sensitivity and specificity at optimal Youden index are displayed for each ROC curve.

desirable for some classification tasks, notably for inflammation grade, but we did not pursue this task. As the number of hyper-parameters to be tuned for SVM is larger than for RF, further tuning of the SVM model might require a grid search on a validation set in the case of a larger sample size.

**Ordinal relation between steatosis, inflammation or fibrosis and QUS/pSWE features.** Ordinal logistic regressions revealed significant ordinal relations (adopting a confidence level of 0.05) between steatosis grade and mean value (within the segmented ROI) of the

normalized mean intensity $\mu_n$ ($p = 0.0039$), mean value and IQR of the coherent-to-diffuse signal ratio $k$ ($p = 0.0029$ and $0.029$, respectively), and the local ACS ($p < 0.001$). There was also a significant relation between the fibrosis stage and shear wave elasticity ($p = 0.0011$). See Table 3 for standard deviations of regression coefficients.

### Categorical association between steatosis, inflammation, and fibrosis

There was an association between inflammation grade and fibrosis stage ($p = 0.010$) but not between steatosis and inflammation grades ($p = 0.57$), nor between steatosis grade and fibrosis stage ($p = 0.97$).

### Discussion

This study revealed that a machine learning approach combining QUS features with elasticity provides a higher accuracy than elasticity alone for the classification of steatosis, inflammation, and fibrosis in patients with CLD. Ultrasound-based elastography techniques provide high accuracy for staging liver fibrosis [44, 45]. However, performance is limited in the case of steatosis and inflammation, which may coexist with fibrosis and confound liver stiffness [21, 46]. It has been reported that liver steatosis decreases liver stiffness in mixed steatohepatitis condition [6, 46, 47], and that inflammation, through a combination of hepatocellular ballooning, aggregation of macrophages, and edema, increases liver stiffness [6, 48–51].

Steatosis grades were best classified by a combination of QUS parameters (including coherent-to-diffuse signal ratio, local attenuation coefficient slope, mean intensity normalized by its maximal value, and shear wave elasticity). Multi-parametric approaches provided good to excellent classification accuracy: 0.90 for steatosis grades S0 vs. 1–3, 0.81 for grades S0-1 vs. S2-3, and 0.78 for grades S0-2 vs. S3. Based on coefficients of proportional odds logistic regressions, the local ACS tended to increase with steatosis grade, as expected to occur with an increase in fat content, in agreement with Goshal et al. [12]. Related to the increase in local ACS, a trend of decrease in normalized intensity ($\mu_n$) was observed. On the other hand, the backscatter coefficient, which is related to the intensity itself ($\mu$), has been reported to be positively correlated with steatosis grade [52]. Moreover, as noted in Zhou et al. [22], an increase in fat content (*i.e.*, fat-infiltrated hepatocytes with nuclei pushed to the cell periphery) is

**Table 3. Coefficients (with 95% confidence intervals in parentheses) of ordinal logistic regressions with the steatosis grade, or the inflammation grade, or the fibrosis stage as responses, and with features appearing in Table 1, as single predictors.**

| Pathological features | pSWE | $\mu_n$ Mean | $\mu_n$ IQR | $1/\alpha$ IQR | $k$ Mean | $k$ IQR | $1/(k+1)$ Mean | $1/(k+1)$ IQR | Total ACS. | Local ACS. |
|---|---|---|---|---|---|---|---|---|---|---|
| Steatosis | -0.80 (-1.44, -0.23) | **-0.033 (-0.052, -0.016)** | — | — | **2.84 (1.46, 4.47)** | **1.65 (0.67, 2.83)** | — | -5.39 (-25.13, 12.90) | — | **9.75 (6.31, 13.71)** |
| | $p = 0.097$ | **$p = 0.0039$** | | | **$p = 0.0027$** | **$p = 0.029$** | | $p = 1.0$ | | **$p < 0.001$** |
| Inflammation | 0.82 (0.23, 1.43) | — | — | -11.12 (-24.54, 1.95) | -0.95 (-2.12, 0.20) | — | — | — | -0.28 (-0.78, 0.20) | — |
| | $p = 0.083$ | | | $p = 0.76$ | $p = 0.75$ | | | | $p = 1.0$ | |
| Fibrosis | **1.50 (0.82, 2.29)** | — | -0.0009 (-0.0112, 0.0095) | — | -1.08 (-2.24, 0.01) | -0.57 (-1.42, 0.25) | 7.99 (0.59, 15.93) | 9.90 (-6.78, 28.36) | -0.05 (-0.52, 0.40) | — |
| | **$p = 0.0011$** | | $p = 0.86$ | | $p = 0.51$ | $p = 1.0$ | $p = 0.39$ | $p = 1.0$ | $p = 1.0$ | |

ACS = attenuation coefficient slope. AUC-ROC = area under the receiver operating characteristic curve. Numbers in parentheses are 95% confidence intervals. pSWE = point shear wave elasticity; $\mu_n$ = mean intensity normalized by its maximal value; $1/\alpha$ = reciprocal of the scatterer clustering parameter; $k$ = coherent-to-diffuse signal ratio; $1/(k+1)$ = diffuse-to-total signal power ratio; IQR = inter-quartile range. Holm-Bonferroni correction was applied to $p$-values. A positive coefficient indicates an increasing ordinal relation between the predictor and the response.

expected to yield an increase in the coherent component of the analytic ultrasound signal. The observed trend was indeed an increase in the average coherent-to-diffuse signal ratio with steatosis grade. There was also a corresponding trend for the IQR of this parameter. Features that did not perform well as stand-alone features could nonetheless improve classification performance when combined with other features, although this makes physical interpretation of their contribution more difficult to analyze. Our proposed method compares favorably with recent results reported by Moret et al. for two state-of-the-art techniques: hepatorenal index ratio with AUCs of 0.90 for S0 vs. S1-3, 0.78 for S0-1 vs. S2-3, and 0.73 for S0-2 vs. S3 and controlled attenuation parameter (CAP) with AUCs of 0.93 for S0 vs. S1-3, 0.76 for S0-1 vs. S2-3, and 0.70 for S0-2 vs. S3 [53].

Inflammation grades were best classified by QUS and elastography features (including total attenuation coefficient slope, coherent-to-diffuse signal ratio, point shear wave elasticity, and reciprocal of the scatterer clustering parameter). Multi-parametric approaches provided AUCs ranging from 0.68 to 0.75, significantly higher than elastography alone, which provided AUCs ranging from 0.56 to 0.64. There was no significant ordinal relation between inflammation grade and any features retained in the 3 binary inflammation classification tasks. However, pSWE was somewhat related to inflammation grade with a *P* value of 0.083. This trend of increase in stiffness with inflammation is consistent with reported results in an animal study [47].

Fibrosis staging with pSWE alone provided rather low performance for classification of fibrosis stages F0 vs. F1-4 and moderate performance ranging from 0.72 to 0.77 for fibrosis stages F0-1 vs. F2-4, F0-2 vs. F3-4, and F0-3 vs. 4. We believe that the coexistence of confounding factors in a heterogeneous disease population may explain the lower AUCs of pSWE in detecting advanced fibrosis, whereas many prior studies on elastography focused on populations with only one disease. Nonetheless, the combination with QUS parameters (coherent-to-diffuse signal ratio, diffuse-to-total signal power ratio, mean intensity normalized by its maximal value, and total attenuation) could further improve the classification accuracy, leading to AUCs in the range of 0.72 to 0.77 for all dichotomization schemes. Among features that were retained for the 4 binary fibrosis classification tasks, pSWE presented an ordinal relation with fibrosis stage. The observed trend of increase in stiffness with fibrosis stage is consistent with the multivariate analysis reported in Kazemirad et al. [47]. Recent works by Brattain et al, Durot et al., Kagadis et al., showed that fibrosis classification accuracy could be further improved, with AUCs in the range of 0.93 to 0.99, by using machine learning techniques such as automated image quality assessment and ROI selection [54], support vector machines [16] and deep learning techniques applied on shear wave elastography data [19].

These results confirm that using machine learning to select the best combination of QUS and elastography parameters may significantly improve classification of steatosis, inflammation, and fibrosis in patients with CLD in comparison with elastography alone. Such a comprehensive approach is critical to reduce or alleviate the need for liver biopsy because these histological components coexist and may have confounding effects on liver stiffness. Of note, our study found associations between inflammation grade and fibrosis stage. Therefore, a multi-parametric approach was required to account for the stiffness-increasing effect of inflammation and fibrosis.

Notice that for some of the classification tasks, distinct combinations of features yielded the same AUC, up to two decimals: steatosis grade S0 vs. S1-3, with QUS features combined with pSWE; steatosis grade S0-2 vs. S3, also with QUS features only; fibrosis stage F0-1 vs. $\geq$ F2-4, fibrosis stage F0-2 vs. F3-4 and fibrosis stage F0-3 vs. F4, also with QUS features combined with pSWE. In particular, in all cases where the best combination of features comprised pSWE, none of the tested combinations of QUS features only were as good.

## Limitations

The sample size was limited to 82 patients with various causes of CLD. However, this pilot study constitutes a proof of concept indicating that incorporating QUS and elastography increases the classification study, a line of work that should be investigated in larger studies with a single etiology such as NAFLD. We did not assess iron overload, another histological feature encountered in some causes of CLD. While there is currently no ultrasound-based technique for assessment of iron overload, this remains a relatively uncommon clinical problem encountered in select populations with hemochromatosis and transfusional hemosiderosis that can be detected by blood tests and quantitated by magnetic resonance imaging [55]. Another limitation was the lack of assessment of biliary disease. Similarly, biliary disease can be suspected on the basis of cholestatic enzymes and assessed by magnetic resonance cholangiopancreatography when relevant. Ultrasound images were acquired with two clinical scanner models which could have introduced variability. Finally, our dataset was imbalanced with few positive cases for steatosis grades S0-2 vs. S3 (66 vs. 16) and inflammation grades A0-2 vs. A3 (74 vs. 8), and with few negative cases for inflammation grades A0 vs. A1-3 (8 vs. 74) and fibrosis stages F0 vs. F1-4 (12 vs. 70). However, we have used the *G*-mean as metric for features selection, which is recommended in the case of imbalanced datasets. Moreover, in that context, AUC under the Precision-Recall curve (PRC) has been suggested as alternative metric to AUC-ROC [56]. We have verified that AUC-PRCs were higher for combinations of features than with pSWE stand-alone, albeit further assessment in future studies would require larger sample size. Further assessment would be required on a larger data set in a target population of patients with NAFLD or NASH, in particular to assess more precisely the impact of each of the proposed QUS features in the considered dichotomous tasks. Future work could include clinical variables, such as body mass index, to take into account variability in patient's health conditions, at the classifier's training step. Furthermore, other machine learning models than random forests could be assessed on a larger database.

## Conclusions

In conclusion, this ancillary study to a prospective imaging trial revealed that a machine learning model based on random forests could select combinations of QUS features and shear wave elastography stiffness that improved the classification accuracy of three key histological features of CLD (steatosis, inflammation, and fibrosis) over that of elastography alone. Future research should validate this approach in human cohorts with NAFLD, a highly prevalent cause of CLD.

## Supporting information

**S1 Table. Accuracy of pSWE and QUS features.** Accuracy of each of the eleven features alone and for the best combination of features for classification of steatosis, inflammation, and fibrosis. The best combination depends on the classification task; see Table 2.
(DOCX)

**S2 Table. ANOVA tests and post-hoc multiple comparisons when applicable for each of these features.** Anova (AOV) test if Shapiro-Wilk (S-W) test succeeded (*p*-value > 1) or else Kruskal-Wallis (K-S) tests with post-hoc multiple comparison tests with Bonferroni-Holm adjustment of *p*-values, based on t-test in the former case, or Wilcoxon rank sum test in the latter case.
(DOCX)

**S3 Table. Accuracy of all eleven features obtained with 0.638+ bootstrap method.** All features taken as input to random forests and of the best combination of features for classification of steatosis, inflammation, and fibrosis. The best combination of features depends on the classification task; see Table 2.
(DOCX)

**S4 Table. Dataset.** Dataset contains patient identification from 1 to 82 (ID), steatosis grade (Steatosis), inflammation grade (Inflammation), fibrosis stage (Fibrosis), point shear wave elasticity (pSWE), $\mu_n$ = mean intensity normalized by its maximal value (munMean), $1/\alpha$ = reciprocal of the scatterer clustering parameter (ialphaMean), $k$ = coherent-to-diffuse signal ratio (kMean), $1/(k + 1)$ = diffuse-to-total signal power ratio (ikappaMean), mean intensity normalized by its maximal value inter-quartile range (munIQR), reciprocal of the scatterer clustering parameter inter-quartile range (ialphaIQR), coherent-to-diffuse signal ratio inter-quartile range (kIQR), diffuse-to-total signal power ratio inter-quartile range (ikappaIQR), total attenuation coefficient slope (TotalACS), local attenuation coefficient slope (LocalACS).
(XLSX)

**S5 Table. Importance of each parameter within the multi-parameter combinations appearing in Table 2 for the various dichotomous classification tasks.** The reported importance is represented by the mean decrease in accuracy, where a higher value corresponds to a higher importance.
(DOCX)

**S6 Table. Accuracy of shear wave elasticity alone and in combination with quantitative ultrasound (QUS) features for classification of steatosis, inflammation, and fibrosis, based on support vector machine (SVM) models.** The same combinations of parameters as in Table 2 were tested with SVM. The parameter γ of radial basis functions (RBF) was set to the reciprocal of the number of parameters (default value with R package "e1071"); the degree of RBFs was the default value 3; the cost $C$ was varied according to powers of 2 for each combination of parameters and the best resulting AUC-ROC is reported.
(DOCX)

## Acknowledgments

We thank Mrs. Assia Belblidia, Mrs. Catherine Huet, and Mr. Walid El Abyad for their assistance in patient enrollment. ACUSON S2000 and S3000 ultrasound systems were lent by Siemens Healthineers.

## Author Contributions

**Conceptualization:** Giada Sebastiani, Guy Cloutier, An Tang.

**Data curation:** Marc Gesnik, Boris Chayer, An Tang.

**Formal analysis:** François Destrempes, Marc Gesnik, Boris Chayer, Marie-Hélène Roy-Cardinal, An Tang.

**Funding acquisition:** Giada Sebastiani, Guy Cloutier, An Tang.

**Investigation:** François Destrempes, Marc Gesnik, Boris Chayer, Marie-Hélène Roy-Cardinal, Jeanne-Marie Giard, Giada Sebastiani, Bich N. Nguyen, Guy Cloutier, An Tang.

**Methodology:** François Destrempes, Marc Gesnik, Boris Chayer, Marie-Hélène Roy-Cardinal, Damien Olivié, Jeanne-Marie Giard, Giada Sebastiani, Bich N. Nguyen, Guy Cloutier, An Tang.

**Project administration:** Damien Olivié, Jeanne-Marie Giard, Giada Sebastiani, Guy Cloutier, An Tang.

**Resources:** Boris Chayer, Damien Olivié, Giada Sebastiani, Bich N. Nguyen, Guy Cloutier, An Tang.

**Software:** François Destrempes, Boris Chayer.

**Supervision:** Damien Olivié, Jeanne-Marie Giard, Giada Sebastiani, Bich N. Nguyen, Guy Cloutier, An Tang.

**Validation:** Jeanne-Marie Giard, Bich N. Nguyen, Guy Cloutier, An Tang.

**Visualization:** François Destrempes, Jeanne-Marie Giard, Guy Cloutier, An Tang.

**Writing – original draft:** François Destrempes, Guy Cloutier, An Tang.

**Writing – review & editing:** François Destrempes, Marc Gesnik, Boris Chayer, Marie-Hélène Roy-Cardinal, Damien Olivié, Jeanne-Marie Giard, Giada Sebastiani, Bich N. Nguyen, Guy Cloutier, An Tang.

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
