## [Decision Letter · Decision Letter 0]

13 Aug 2021

PONE-D-21-21119

Quantitative ultrasound, elastography, and machine learning for assessment of steatosis, inflammation, and fibrosis in chronic liver disease

PLOS ONE

Dear Dr. Tang,

Thank you for submitting your manuscript to PLOS ONE. After careful consideration, we feel that it has merit but does not fully meet PLOS ONE’s publication criteria as it currently stands. Therefore, we invite you to submit a revised version of the manuscript that addresses the points raised during the review process.

We look forward to receiving your revised manuscript.

Kind regards,

Omar Sultan Al-Kadi, PhD

Academic Editor

PLOS ONE

 Journal Requirements: When submitting your revision, we need you to address these additional requirements. 1. Please ensure that your manuscript meets PLOS ONE's style requirements, including those for file naming. The PLOS ONE style templates can be found at https://journals.plos.org/plosone/s/file?id=wjVg/PLOSOne_formatting_sample_main_body.pdf and https://journals.plos.org/plosone/s/file?id=ba62/PLOSOne_formatting_sample_title_authors_affiliations.pdf 2. Thank you for stating the following in the Acknowledgments Section of your manuscript: [We thank Mrs. Assia Belblidia, Mrs. Catherine Huet, and Mr. Walid El Abyad for their assistance in patient enrollment. ACUSON S2000 and S3000 ultrasound systems were lent by Siemens Healthineers. This work was supported by grants from the Canadian Institutes of Health Research (CIHR)-Institute of Nutrition, Metabolism and Diabetes (INMD) (CIHR-INMD #273738 and #301520) and Fonds de recherche du Québec en Santé (FRQS) and Fondation de l'Association des radiologistes du Québec (FARQ) Clinical Research Scholarship – Junior 1 and 2 Salary Award (FRQS-FARQ #26993 and #34939) to An Tang. Junior 1 Salary Award from FRQS (#27127) and research salary from McGill University to Giada Sebastiani.] We note that you have provided funding information that is not currently declared in your Funding Statement. However, funding information should not appear in the Acknowledgments section or other areas of your manuscript. We will only publish funding information present in the Funding Statement section of the online submission form. Please remove any funding-related text from the manuscript and let us know how you would like to update your Funding Statement. Currently, your Funding Statement reads as follows:  [This work was supported by grants from the Canadian Institutes of Health Research (CIHR)-Institute of Nutrition, Metabolism and Diabetes (INMD) (CIHR-INMD #273738 and #301520, https://cihr-irsc.gc.ca/) to AT.  This work was also supported by Junior 1 and Junior 2 Clinical Research Scholarships from the Fonds de recherche du Québec en Santé (FRQS, https://frq.gouv.qc.ca/en/) and Fondation de l'Association des radiologistes du Québec (FARQ) (FRQS-FARQ #26993 and #34939 to AT; and by a Junior 1 Clinical Research Scholarships from FRQS (#27127) and research salary from McGill University to GS.] Please include your amended statements within your cover letter; we will change the online submission form on your behalf. 3. Thank you for stating the following in your Competing Interests section:  [No]. Please complete your Competing Interests on the online submission form to state any Competing Interests. If you have no competing interests, please state "The authors have declared that no competing interests exist.", as detailed online in our guide for authors at http://journals.plos.org/plosone/s/submit-now  This information should be included in your cover letter; we will change the online submission form on your behalf. 4. Please include captions for your Supporting Information files at the end of your manuscript, and update any in-text citations to match accordingly. Please see our Supporting Information guidelines for more information: http://journals.plos.org/plosone/s/supporting-information. 

Reviewers' comments:

Reviewer's Responses to Questions

**Comments to the Author**

1. Is the manuscript technically sound, and do the data support the conclusions?

Reviewer #1: Yes

Reviewer #2: Partly

Reviewer #3: Yes

2. Has the statistical analysis been performed appropriately and rigorously? 

Reviewer #1: Yes

Reviewer #2: Yes

Reviewer #3: Yes

3. Have the authors made all data underlying the findings in their manuscript fully available?

Reviewer #1: No

Reviewer #2: No

Reviewer #3: No

4. Is the manuscript presented in an intelligible fashion and written in standard English?

Reviewer #1: Yes

Reviewer #2: Yes

Reviewer #3: Yes

5. Review Comments to the Author

Reviewer #1: A very nicely structured paper. I have some suggestions that may help the authors to improve the paper.

--Lines 186-192. It's not clear how these parameters were generated. I would suggest clarifying this section

-- NASH-CRN system uses 1a, 1b and 1c. Did you pool these values as 1 and merged with the rest of the group? (METAVIR for other etiologies).

--Using two different ultrasound models (S2000 and S3000) could cause variability. I would suggest adding this factor to the limitations.

--I would suggest annotating Fig3 and showing the areas with fibrosis, inflammation and steatosis.

--I would suggest providing some images from the pSWE exams.

--I would suggest using NAS scores and performing analysis if available.

Thank you for considering my recommendations.

Reviewer #2: General Comments:

This study presents a machine learning method for assessing chronic liver disease (CLD) diagnostic parameters (fibrosis, steatosis, inflammation) through ultrasound (US) B-Mode and Elastographic information provided by ultrasound imaging. The study has some merits as the combination of information of fibrosis, steatosis and inflammation for CLD assessment has not yet been evaluated. Also, the statistical analysis of the Results provided is good and properly written. Its novelty is limited to this combination though, as the algorithms used are already known and used widely in the literature. Its flow is sometimes confusing as many aspects referred for the first time are explained in other, later parts of the manuscript. Furthermore, the manuscript has some major flaws that should be considered before publication.

Non-Specific Comments:

1. The study’s text flow is sometimes confusing as many aspects referred for the first time are explained in other, later parts of the manuscript. For example, the histological information section containing Metavir Classification systems for fibrosis and inflammation (activity) should be earlier in the manuscript as there are points that stage-grade information is shown that are unknown to the non-expert reader, and are not explained. For a large part of the manuscript it is not clear whether the Metavir or Ishak classification system is used. Also, Table 2 should be transferred in the Results section as it is early in the Methods section.

2. The stage-grade notation is a little confusing as usually, in the literature, there is a capital letter ahead of stage-grade number e.g. F≥F2 for fibrosis, S≥S1 for steatosis and A≥A3 for inflammation. Please provide clearer stage-grade class representation.

3. The Introduction section needs many additions regarding existing literature. Only clinical information of the CLD is mentioned and minimal information is given on the non-invasive approaches (clinical or A.I. based) for CLD assessment. Many elastographic variants exist (MR- and US-based) that present variable performances in clinical and A.I. related studies that are completely absent on this study. Also, no information is given on other A.I. related works on CLD assessment. What is their performance? Their performance should be discussed in comparison with the literature.

a. Indicative works (much more exist) that the Authors should include and discuss are:

i. Gatos, I., Tsantis, S., et al, (2016), A new computer aided diagnosis system for evaluation of chronic liver disease with ultrasound shear wave elastography imaging. Med. Phys., 43: 1428–1436. doi:10.1118/1.4942383

ii. Gatos, I., Tsantis, S., et al, (2017), A Machine-Learning Algorithm Toward Color Analysis for Chronic Liver Disease Classification, Employing Ultrasound Shear Wave Elastography, Ultrasound in Medicine and Biology, Volume 43, Issue 9, 1797 – 1810. doi:10.1016/j.ultrasmedbio.2017.05.002

iii. Gatos, I., Tsantis, S., et al, (2019), Temporal stability assessment in shear wave elasticity images validated by deep learning neural network for chronic liver disease fibrosis stage assessment. Med. Phys., 46(5): 2298-2309. doi:10.1002/mp.13521

iv. Gatos, I., Drazinos, P., et al, (2020), Comparison of Sound Touch Elastography, Shear Wave Elastography and Vibration-Controlled Transient Elastography in Chronic Liver Disease Assessment using Liver Biopsy as the "Reference Standard". Ultrasound in Medicine and Biology, 46(4):959-971. doi:10.1016/j.ultrasmedbio.2019.12.016.

v. Kagadis, G.C., Drazinos, P., et al, (2020), Deep learning networks on chronic liver disease assessment with fine-tuning of shear wave elastography image sequences. Phys. Med. Biol. 65 215027

vi. Stoean R, Stoean C, Lupsor M, Stefanescu H and Badea R 2011 Evolutionary-driven support vector machines for determining the degree of liver fibrosis in chronic hepatitis C Artif. Intell. Med. 51 53–65

vii. Wang K et al 2019 Deep learning radiomics of shear wave elastography significantly improved diagnostic performance for assessing liver fibrosis in chronic hepatitis B: a prospective multicentre study Gut 68 729–41

viii. Meng D, Zhang L, Cao G, Cao W, Zhang G and Hu B 2017 Liver fibrosis classification based on transfer learning and FCNet for ultrasound images IEEE Access 5 5804–10

ix. Durot I, Akhbardeh A, Sagreiya H, Loening A M and Rubin D L 2020 A new multimodel machine learning framework to improve hepatic fibrosis grading using ultrasound elastography systems from different vendors Ultrasound Med. Biol. 46 26–33

4. The Methods section has also issues as:

a. The Authors compare plain pSWE measurement with all the information processed in the proposed machine learning scheme that is somewhat irrelevant or, at least, should be just complementary. The authors should compare the proposed algorithm’s estimation with the combination of information given by the pSWE measurement and the US B-Mode image by an expert clinician or by other analysis.

b. Only binary class differentiation is presented. Although this is not a major limitation as other class combinations (e.g. ternary or full) may be evaluated due to small sample in certain classes, some could be realized as they may contain significant clinical information (e.g. Mild Fibrosis vs. Significant Fibrosis vs. Cirrhosis, or in other words, F0-F1 vs. F2-F3 vs. F4).

c. No splitting of data (e.g. 70% training to 30% validation) is provided resulting in no estimation of the robustness of the results.

5. The Discussion section is incomplete in terms of comparison with relevant clinical and technical works in the literature that is also absent in the Introduction. That being said, the proposed method shows relatively low performance scores (~0.70-0.80) for fibrosis, steatosis and inflammation. Compared to the literature for fibrosis, these are much lower, especially for the proposed scheme to other machine learning schemes. If the proposed scheme’s performance on CLD assessment is inferior to other A.I. variants processing US imaging information, why should it be published? Regarding Steatosis comparison with other popular parameters in the literature as the Hepatorenal Index (HRI) or the Controlled Attenuation Parameter (CAP) is absent. Regarding inflammation, other studies have discussed that elastography in general, provides no significant clinical information. Is this study’s result a confirmation on this aspect? Please discuss and compare with literature. Also, as there are studies in the literature having deep learning implementations on visualized elastography (e.g. Supersonic’s SWE) with far better results, why should a machine learning scheme such as this be preferred? Please also discuss.

Specific Comments:

1. Page 9, Abstract: “This ancillary study to a prospective institutional review-board approved study included 82 patients with non-alcoholic fatty liver disease, chronic hepatitis B or C virus, or autoimmune hepatitis.”

Bad English here. Also, there are some aspects that are unclear. For example, the use of the word «prospective» when in the Methods section you mention that this is a retrospective study. Also, i don’t understand the «ancillary» part that this study fulfills and is mentioned also in the Methods section. Please explain.

Reviewer #3: This paper presented a machine learning method that used QUS and pSWE to classify steatosis grade, inflammation grade, and fibrosis stage in the CLD patient population. Histopathology was used as the gold standard. Results indicated that the addition of QUS was superior than using SWE alone.

The paper was clearly written and easy to follow. The topic is of interest to the medical ultrasound community.

A few comments below:

1. One major weakness of the paper is that only Random Forrest (RF) was used. There is no other machine learning method to compare the RF results against to. It’s well known in the SWE community that due to the current challenge of high variability in SWE acquisition and measurement, SWE alone is not a reliable measurement for inflammation, fibrosis and steatosis grading. Due to the reason above, it is almost expected that combining QUS and SWE will result in improved assessment, therefore without additional method(s) to compare against it is difficult to understand the efficacy of the RF method presented here.

Recommend adding at least one additional machine learning method.

2. As we can see from Table 2, the parameters chosen for each of the binary classification tasks vary. This begs the question of how practical this RF-based method is. Furthermore, it’s perceivable that with addition of new data in the future, the parameters will change, which means that the results presented here are far from conclusive.

3. Minor comment: some of the text in the manuscript is in bold. Unclear if it follows any particular pattern. Recommend removing or cleaning up the bold formatting.

6. PLOS authors have the option to publish the peer review history of their article (what does this mean?). If published, this will include your full peer review and any attached files.

Reviewer #1: No

Reviewer #2: No

Reviewer #3: No

---

## [Author Response · Author response to Decision Letter 0]

20 Sep 2021

Please refer to the Response to Reviewers.docx document which is formatted to facilitate differentiation between reviewers' comments, responses, and modifications to the manuscript, figures, and supporting information.

---

## [Decision Letter · Decision Letter 1]

26 Oct 2021

PONE-D-21-21119R1Quantitative ultrasound, elastography, and machine learning for assessment of steatosis, inflammation, and fibrosis in chronic liver diseasePLOS ONE

Dear Dr. Tang,

Thank you for submitting your manuscript to PLOS ONE. After careful consideration, we feel that it has merit but does not fully meet PLOS ONE’s publication criteria as it currently stands. Therefore, we invite you to submit a revised version of the manuscript that addresses the points raised during the review process.

We look forward to receiving your revised manuscript.

Kind regards,

Omar Sultan Al-Kadi, PhD

Academic Editor

PLOS ONE

Journal Requirements:

Reviewers' comments:

Reviewer's Responses to Questions

**Comments to the Author**

1. If the authors have adequately addressed your comments raised in a previous round of review and you feel that this manuscript is now acceptable for publication, you may indicate that here to bypass the “Comments to the Author” section, enter your conflict of interest statement in the “Confidential to Editor” section, and submit your "Accept" recommendation.

Reviewer #1: All comments have been addressed

Reviewer #2: All comments have been addressed

Reviewer #3: All comments have been addressed

2. Is the manuscript technically sound, and do the data support the conclusions?

Reviewer #1: Yes

Reviewer #2: Yes

Reviewer #3: Yes

3. Has the statistical analysis been performed appropriately and rigorously? 

Reviewer #1: Yes

Reviewer #2: Yes

Reviewer #3: Yes

4. Have the authors made all data underlying the findings in their manuscript fully available?

Reviewer #1: Yes

Reviewer #2: Yes

Reviewer #3: Yes

5. Is the manuscript presented in an intelligible fashion and written in standard English?

Reviewer #1: Yes

Reviewer #2: Yes

Reviewer #3: Yes

6. Review Comments to the Author

Reviewer #1: Thank you for addressing my comments. I do not have any further suggestions. Congratulations

As a minor note. The authors may consider adding this paper to their discussion as well (PMID: 32622685)

Reviewer #2: (No Response)

Reviewer #3: This paper presented an ML-based method, specifically Random Forest (RF), to classify three important grades for the CLD population: steatosis, inflammation, and fibrosis. Feature sets of QUS and pSWE combind were compared against pSWE alone. When tested on a dataset of 82 patient, RF on combined feature sets outperformed pSWE alone.

The paper is clearly written and easy to follow. The dataset description is thorough, the experiment setup is clear, the results can be easily understood.

Major comments:

1. The selected feature set is different for each classification tasks. What are the overlapping features among these classification tasks?

2. RF can also rank the weights of the selected features. Which features carry more weights? This is helpful to understand, because it can help interpret the ML results further.

3. One of the biggest limitations of the paper, as mentioned in the discussion, is that only RF is presented. SVM has been reported by other papers, it would be helpful to see how the RF classifier compares against SVM given the feature sets used.

Minor comments:

1. The font size is different at a number of locations, e.g., Line 279, 281, 302, 343, 526, etc.

2. Figures appear blurry, but that could be the result of how the pdf is generated.

7. PLOS authors have the option to publish the peer review history of their article (what does this mean?). If published, this will include your full peer review and any attached files.

Reviewer #1: No

Reviewer #2: No

Reviewer #3: No

---

## [Author Response · Author response to Decision Letter 1]

8 Dec 2021

Please refer to the file entitled Response to Reviewers.docx

---

## [Decision Letter · Decision Letter 2]

21 Dec 2021

Quantitative ultrasound, elastography, and machine learning for assessment of steatosis, inflammation, and fibrosis in chronic liver disease

PONE-D-21-21119R2

Dear Dr. Tang,

We’re pleased to inform you that your manuscript has been judged scientifically suitable for publication and will be formally accepted for publication.

Kind regards,

Omar Sultan Al-Kadi, PhD

Academic Editor

PLOS ONE

Additional Editor Comments (optional):

Reviewers' comments:

Reviewer's Responses to Questions

**Comments to the Author**

1. If the authors have adequately addressed your comments raised in a previous round of review and you feel that this manuscript is now acceptable for publication, you may indicate that here to bypass the “Comments to the Author” section, enter your conflict of interest statement in the “Confidential to Editor” section, and submit your "Accept" recommendation.

Reviewer #1: All comments have been addressed

2. Is the manuscript technically sound, and do the data support the conclusions?

Reviewer #1: Yes

3. Has the statistical analysis been performed appropriately and rigorously? 

Reviewer #1: Yes

4. Have the authors made all data underlying the findings in their manuscript fully available?

Reviewer #1: Yes

5. Is the manuscript presented in an intelligible fashion and written in standard English?

Reviewer #1: Yes

6. Review Comments to the Author

Reviewer #1: Thank you for addressing my comments. The manuscript looks great.

I do not have any further suggestions. Thank you.

7. PLOS authors have the option to publish the peer review history of their article (what does this mean?). If published, this will include your full peer review and any attached files.

Reviewer #1: No

---

## [Editor Report · Acceptance letter]

14 Jan 2022

PONE-D-21-21119R2 

Quantitative ultrasound, elastography, and machine learning for assessment of steatosis, inflammation, and fibrosis in chronic liver disease 

Dear Dr. Tang:

I'm pleased to inform you that your manuscript has been deemed suitable for publication in PLOS ONE. Congratulations! Your manuscript is now with our production department. 

Kind regards, 

on behalf of

Dr. Omar Sultan Al-Kadi 

Academic Editor

PLOS ONE